# Stereoselective Synthesis and Anticancer Activity of 2,6-Disubstituted *trans*-3-Methylidenetetrahydropyran-4-ones

**DOI:** 10.3390/ma15093030

**Published:** 2022-04-21

**Authors:** Tomasz Bartosik, Joanna Drogosz-Stachowicz, Anna Janecka, Jacek Kędzia, Barbara Pacholczyk-Sienicka, Jacek Szymański, Katarzyna Gach-Janczak, Tomasz Janecki

**Affiliations:** 1Institute of Organic Chemistry, Lodz University of Technology, Żeromskiego 116, 90-924 Lodz, Poland; tomasz.bartosik@p.lodz.pl (T.B.); jacek.kedzia@p.lodz.pl (J.K.); barbara.pacholczyk@p.lodz.pl (B.P.-S.); 2Department of Biomolecular Chemistry, Medical University of Łódź, Mazowiecka 6/8, 92-215 Lodz, Poland; joanna.drogosz-stachowicz@stud.umed.lodz.pl (J.D.-S.); anna.janecka@umed.lodz.pl (A.J.); katarzyna.gach-janczak@umed.lodz.pl (K.G.-J.); 3Central Scientific Laboratory, Medical University of Łódź, Mazowiecka 6/8, 92-215 Lodz, Poland; jacek.szymanski@umed.lodz.pl

**Keywords:** 3-methylidenedihydropyran-4-ones, Michael addition, Horner–Wadsworth–Emmons olefination, cytotoxic activity, apoptosis, cell cycle, topoisomerase II inhibitors

## Abstract

In this report, we present efficient and stereoselective syntheses of 2,6-disubstituted *trans*-3-methylidenetetrahydropyran-4-ones and 2-(4-methoxyphenyl)-5-methylidenetetrahydropyran-4-one that significantly broaden the spectrum of the available methylidenetetrahydropyran-4-ones with various substitution patterns. Target compounds were obtained using Horner–Wadsworth–Emmons methodology for the introduction of methylidene group onto the pyranone ring. 3-Diethoxyphosphoryltetrahydropyran-4-ones, which were key intermediates in this synthesis, were prepared by fully or highly stereoselective addition of Gilman or Grignard reagents to 3-diethoxyphosphoryldihydropyran-4-ones. Addition occurred preferentially by axial attack of the Michael donors on the dihydropyranone ring. Relative configurations and conformations of the obtained adducts were assigned using a detailed analysis of the NMR spectra. The obtained methylidenepyran-4-ones were evaluated for cytotoxic activity against two cancer cell lines (HL-60 and MCF-7). 2,6-Disubstituted 3-methylidenetetrahydropyran-4-ones with isopropyl and phenyl substituents in position 2 were more cytotoxic than analogs with *n-*butyl substituent. Two of the most cytotoxic analogs were then selected for further investigation on the HL-60 cell line. Both analogs induced morphological changes characteristic of apoptosis in cancer cells, significantly inhibited proliferation and induced apoptotic cell death. Both compounds also generated DNA damage, and one of the analogs arrested the cell cycle of HL-60 cells in the G2/M phase. In addition, both analogs were able to inhibit the activity of topoisomerase IIα. Based on these findings, the investigated analogs may be further optimized for the development of new and effective topoisomerase II inhibitors.

## 1. Introduction

The tetrahydropyran skeleton is present in a great number of natural products, which show various important biological properties, including anticancer activity. This activity depends on many structural and stereochemical features, and one of the most important is the substitution pattern of the tetrahydropyran ring. In this respect, many compounds containing a *trans*-2,6-disubstituted tetrahydropyrane moiety show significant cytotoxic activity. Representative examples are irciniastatins A and B **1** isolated from the Indo-Pacific marine sponge *Ircinia ramosoa*, which are powerful murine and human cancer cell growth inhibitors [1]; and aspergillide C **2** isolated from marine-derived fungus *Aspergillus ostianus* strain 01F313, which showed cytotoxic activity against mouse lymphocytic leukemia cells (L1210) [2] (Figure 1). Another important feature, which can strongly influence cytotoxic activity of tetrahydropyranes is the presence of *exo*-methylidene bond vicinal to the carbonyl group. This moiety is a strong Michael acceptor and can react with various bionucleophiles—in particular, with mercapto groups in cysteine residues of enzymes or other functional proteins. The *exo*-methylidene motif is characteristic of a large group of natural products, including sesquiterpene lactones (e.g., vernolepin **3**) and is believed to be the reason for their strong and often very specific anticancer properties [3,4,5,6]. Several natural compounds containing the 2,6-disubstituted-3-methylidenetetrahydropyran-4-one skeleton are known. Representative examples are norperovskatone **4**, recently isolated from *Perovskia atriplicifolia,* which possesses remarkable anti-hepatitis B virus (HBV) activity [7] or (+)-okilactomycin **5 [8]** which exhibits in vitro cytotoxicity against a number of human cancer cell lines, including lymphoid leukemia L1210 and leukemia P388, and in vivo activity against Ehrlich ascites carcinoma.

Very recently, we reported on the synthesis and anticancer properties of a series of 2,2,6-trisubstituted 5-methylidenetetrahydropyran-4-ones (**6**) [9]. The analysis of the structure–activity relationship of the obtained compounds has revealed that cytotoxicity strongly depends on the nature of the substituent in position 6 (vicinal to the methylidene bond)—butyl and particularly *iso*-propyl groups being the best. The nature of substituents in position 2 is less important, although, in general, aromatic substituents are more advantageous for cancer growth inhibition than non-aromatic ones. One of the obtained compounds, 6-isopropyl-2,2-dimethyl-5-methylidenetetrahydropyran-4-one, showed very high cytotoxic activity against HL-60 human leukemia cells, induced apoptosis of these cells and caused the arrest of the cell cycle in the G2/M phase. Encouraged by these results, in this paper we present a simple, effective and stereoselective synthesis of 2,6-disubstituted *trans*-3-methylidenetetrahydropyran-4-ones **13a–o**, which significantly broadens the spectrum of the available methylidenetetrahydropyran-4-ones with various substitution patterns. We applied synthetic methodology which proved its effectiveness in the synthesis of 5-methylidenetetrahydropyran-4-ones **6** and used Horner–Wadsworth–Emmons reagents for the olefination of formaldehyde to introduce an *exo*-methylidene group onto the tetrahydropyranone ring. In designing substituents for the new methylidenetetrahydropyran-4-one analogs, we turned special attention to these, which proved their effectiveness in enhancing the cytotoxicity of 2,2,6-trisubstituted 3-methylidenetetrahydropyra-4-ones **6**, i.e., isopropyl and *n*-butyl groups. Additionally, we decided to introduce the ferrocenyl moiety onto the tetrahydropyran-4-one skeleton. The ferrocenyl group has proved itself to be an excellent choice to design new drugs [10,11,12], most probably because of its small size, aromaticity, hydrophobicity, low cytotoxicity against normal cells and redox behavior [13,14,15,16,17,18]. Several ferrocene derivatives have already been used as anticancer, antimalarial, antiviral and antibiotic agents [14,15,16,19,20].

The obtained methylidenepyran-4-ones **13a–o** were evaluated for their cytotoxic activity against two cancer cell lines (HL-60 and MCF-7), and for comparison, against a normal cell line (HUVEC). Additionally, the influences of two selected analogs on proliferation, cell cycle distribution, apoptosis induction, DNA damage and topoisomerase IIα activity were also investigated.

## 2. Materials and Methods

### 2.1. Chemistry


**General information**


NMR spectra were recorded on a Bruker DPX 250 or Bruker Avance II Plus instrument (Bruker, Billerica, MA, USA) at 250.13 MHz or 700 for ^1^H; 62.9 or 176 MHz for ^13^C; and 101.3 MHz for ^31^P NMR. Tetramethylsilane was used as an internal standard and 85% H_3_PO_4_ as an external standard. ^31^P NMR spectra were recorded using broadband proton decoupling. Melting points were determined in open capillaries and are uncorrected. Column chromatography was performed on silica gel 60 (230–400 mesh) (Aldrich, Sant Louis, MO, USA). Thin-layer chromatography was performed on the pre-coated TLC sheets of silica gel 60 F254 (Aldrich). The purity of the synthesized compounds was confirmed by the combustion elemental analyses (CH, elemental analyzer EuroVector 3018, Elementar Analysensysteme GmbH, Langenselbold, Germany). MS spectra of intermediates were recorded on Waters 2695-Waters ZQ 2000 LC/MS apparatus. EI mass spectra of final compounds were recorded on a GCMS-QP2010 ULTR A instrument (Shimadzu, Kioto, Japan). The mass spectra were obtained using the following operating conditions: electron energy of 70 eV and ion source temperature of 200 °C. Samples were introduced via a direct insertion probe heated from 30 to 300 °C. All reagents and starting materials were purchased from commercial vendors and used without further purification. Organic solvents were dried and distilled prior to use. Standard syringe techniques were used for transferring dry solvents.


**General procedure for the synthesis of diethyl 4-alkyl(aryl)-4-hydroxy-2-oxobutylphosphonates 9a–e.**


In a round-bottomed three-necked flask under an argon atmosphere, NaH (0.51 g, 17.00 mmol, 80% in mineral oil) was suspended in THF (48 mL). The suspension was stirred, and a solution of diethyl 2-oxopropylphosphonate **7** (3.00 g, 15.45 mmol) in THF (3 mL) was added dropwise. The reaction mixture was stirred for 30 min, cooled below −30 °C in dry ice–acetone bath and n-butyllithium (6.80 mL of 2.5 M solution in hexane, 17.00 mmol) was added dropwise. The reaction mixture was stirred for 30 min at this temperature, cooled to −78 °C in a dry ice–acetone bath and solution of aldehyde (18.54 mmol) in THF (6 mL) was added dropwise. Reaction mixture was stirred for 1.5 h at this temperature. After this time, the reaction was quenched by adding saturated solution of ammonium chloride (100 mL). The water layer was extracted with DCM (3 × 100 mL); the organic layers were combined, washed with brine and dried over MgSO_4_. The solvents were evaporated under reduced pressure, and the resulting crude product was purified by column chromatography (eluent ethyl acetate).

Characteristics of the compounds **9a-e** are given in the Appendix A.


**General procedure for the synthesis of 6-alkyl(aryl)-3-diethoxyphosphoryldihydropyran-4-ones 11a–e.**


The diethyl 4-alkyl(aryl)-4-hydroxy-2-oxobutylphosphonates (**9a–d**) (10.00 mmol) was dissolved in dry dichloromethane (100 mL). Next, dimethyl formamide dimethyl acetal (DMF–DMA) (30.00 mmol, 4.24 mL) was added dropwise and stirred for 2 h. After this time to reaction mixture boron trifluoride etherate (BF_3_ · Et_2_O) (25.00 mmol, 3.10 mL) was added dropwise. In the case of synthesis of 6-alkyl(aryl)-3-diethoxyphosphoryldihydropyran-4-ones **11e**, the reaction was carried without boron trifluoride etherate and reduced amount of dimethyl formamide dimethyl acetal (DMF–DMA) (25.00 mmol, 3.53 mL). Reaction was controlled by ^31^P NMR. After completion of the reaction, to the reaction mixture was added saturated solution of sodium hydrocarbonate (100 mL). The water layer was extracted with ethyl acetate (3 × 100 mL), the organic layers were combined, washed with brine and dried over MgSO_4_. The solvents were evaporated under reduced pressure. Obtained crude product was used on next step without further purification.


**General procedure for the synthesis of 2-alkyl(aryl)-6-alkyl(aryl)-3-diethoxyphosphoryltetrahydropyran-4-ones 12a–j.**


To a suspension of copper iodide (I) (0.80 mmol, 305 mg) in THF (10.70 mL) was added dropwise *n*-BuLi 2.5 M in hexane solution (3.00 mmol, 1.25 mL) at 0 °C. The solution was stirred at this temperature for 20 min. After this time the mixture was cooled to −78 °C, chlorotrimethylsilane (5.00 mmol, 0.65 mL) was added dropwise and stirred for 10 min. Next, to obtained mixture was added a solution of 6-alkyl(aryl)-3-diethoxyphosphoryl- dihydropyran-4-one **11a-e** (1.00 mmol) in dry THF (7.00 mL) and stirred at −78 °C for 4 h. After this time the reaction was quenched with saturated aqueous NH_4_Cl (7.50 mL) and stirred for 20 min. Next, the mixture was diluted with saturated aqueous NH_4_Cl (25.00 mL) and extracted with ethyl acetate (5 × 30 mL). The combined organic layers was washed with H_2_O (50.00 mL) and brine (50.00 mL), then dried over MgSO_4_ and concentrated under reduced pressure. Obtained crude product was purified by column chromatography using as eluent dichloromethane: acetone (20:1) mixture.

Characteristics of the compounds **12a–j** are given in the Appendix A.


**General procedure for the synthesis of 2-isopropyl-6-alkyl(aryl)-3-diethoxyphosphoryltetrahydropyran-4-ones 12k–o.**


The solution of 6-alkyl(aryl)-3-diethoxyphosphoryldihydropyran-4-one **11a-e** (1.0 mmol) in dry THF (10 mL) was cooled to 0 °C in ice-water bath and isopropylmagnesium chloride (3.0 mmol in THF) was added dropwise in argon atmosphere. After 2 h at this temperature, saturated solution of ammonium chloride (15 mL) was added. The water layer was washed with DCM (3 × 15 mL). Combined organic extracts were washed with brine (20 mL) and dried over MgSO_4_. The solvents were evaporated under reduced pressure and the resulting crude product was purified by column chromatography (eluent DCM : Acetone 20:1).

Characteristics of the compounds **12k–o** are given in the Appendix A.

General procedure for the synthesis of 2-alkyl(aryl)-5-diethoxyphosphoryltetrahydropyran-4-ones **14a-d**.

The solution of 6-alkyl(aryl)-3-diethoxyphosphoryldihydropyran-4-one **11a-d** (1.0 mmol) in dry THF (10 mL) was cooled to −78 °C in dry ice–acetone bath and L-selectride^®^ (1.1 mmol in THF) was added dropwise in argon atmosphere. After 1 h at this temperature, the reaction was carried out at 0 °C for 1 h. Reaction was completed by using saturated solution of ammonium chloride (15 mL). The water layer was washed with DCM (3 × 15 mL). Combined organic extracts were washed with brine (20 mL) and dried over MgSO_4_. The solvents were evaporated under reduced pressure and the resulting crude product was purified by column chromatography (eluent DCM : Acetone 20:1).

Characteristics of the compounds **14a-d** are given in the Appendix A.


**General procedure for the synthesis of *trans*-2,6,-disubstituted-3-methylidenetetrahydropyran-4-ones 13a–o and 15d.**


To a vigorously stirred solution of 3-diethoxyphosphoryltetrahydropyran-4-ones **12a–o** and **14d** (0.2 mmol) in dry THF (2 mL), formaldehyde (36–38% solution in water, 0.167 mL, ca. 2.00 mmol) was added at 0 °C, followed by addition of K_2_CO_3_ (55.3 mg, 0.40 mmol) in water (0.56 mL). The resulting mixture was stirred vigorously at 0 °C for 2 h (for pyran-4-ones **12a–o** and **14d**) or at room temperature for 3 h (for pyran-4-ones **12k–o**). Next Et_2_O (10 mL) was added and layers were separated. The water fraction was washed with Et_2_O (5 mL). Organic fractions were combined, washed with brine (10 mL) and dried over MgSO_4_. The solvents were evaporated under reduced pressure and the resulting crude product was purified by column chromatography (eluent dichloromethane: petroleum ether 5:1).

*Trans-6-ethyl-2-isopropyl-3-methylidenetetrahydro-4H-pyran-4-one* (**13a**) (16.0 mg, 44%). Colorless oil. ^1^H NMR (700 MHz, chloroform-*d*) δ 0.88 (d, *J* = 6.6 Hz, 3H), 1.00 (t, *J* = 7.4 Hz, 3H), 1.02 (d, *J* = 6.6 Hz, 3H), 1.54 (dqd, *J* = 13.8, 7.5, 5.0 Hz, 1H), 1.63–1.69 (m, 1H), 1.92 (dhept, *J* = 9.4, 6.6 Hz, 1H), 2.31 (dd, *J* = 16.6, 10.2 Hz, 1H), 2.56 (dd, *J* = 16.6, 3.4 Hz, 1H), 3.85–3.96 (m, 1H), 4.04–4.14 (m, 1H), 5.13 (dd, *J* = 1.3, 1.3 Hz, 1H), 6.00 (dd, *J* = 1.3, 1.3 Hz, 1H). ^13^C NMR (176 MHz, chloroform-*d*) δ 10.02, 18.16, 19.30, 29.33, 29.62, 46.25, 70.36, 82.74, 121.08, 144.16, 198.69. EI-MS [M]^+^ = 182.0. Anal. Calcd for C_11_H_18_O_2_: C, 72.49; H, 9.95. Found: C, 72.28; H, 9.91.

*Trans-2-butyl-6-ethyl-3-methylidenetetrahydro-4H-pyran-4-one* (**13b**) (34.5 mg, 88%). Colorless oil. ^1^H NMR (700 MHz, chloroform-*d*) δ 0.91 (t, *J* = 7.1 Hz, 3H), 0.97 (t, *J* = 7.1 Hz, 3H), 1.20–1.58 (m, 6H), 1.62 (dhept, *J* = 13.7, 7.4 Hz, 1H), 1.81 (dtd, *J* = 13.2, 9.7, 3.9 Hz, 1H), 2.30 (dd, *J* = 16.9, 10.5 Hz, 1H), 2.54 (dd, *J* = 16.9, 3.2 Hz, 1H), 3.88 (dddd, *J* = 10.6, 7.9, 4.9, 3.2 Hz, 1H), 4.59 (dddd, *J* = 10.0, 5.0, 1.5, 1.5 Hz, 1H), 5.17 (dd, *J* = 1.5, 1.5 Hz, 1H), 6.02 (d, *J* = 1.5, 1.5 Hz, 1H). ^13^C NMR (176 MHz, chloroform-*d*) δ 9.94, 14.13, 22.47, 27.68, 29.05, 34.11, 45.86, 69.66, 76.13, 120.26, 145.34, 197.83. EI-MS [M]^+^ = 196.0. Anal. Calcd for C_12_H_20_O_2_: C, 73.43; H, 10.27. Found: C, 73.50; H, 10.26.

*Trans-6-ethyl-3-methylidene-2-phenyltetrahydro-4H-pyran-4-one* (**13c**) (29.0 mg, 67%). Colorless oil. ^1^H NMR (700 MHz, chloroform-*d*) δ 0.90 (t, *J* = 7.4 Hz, 3H), 1.49 (dqd, *J* = 13.8, 7.4, 5.0 Hz, 1H), 1.62 (dp, *J* = 13.8, 7.4 Hz, 1H), 2.39 (dd, *J* = 16.9, 10.6 Hz, 1H), 2.52 (dd, *J* = 16.9, 3.2 Hz, 1H), 3.67 (dddd, *J* = 10.6, 7.9, 5.0, 3.2 Hz, 1H), 5.26 (dd, *J* = 1.3, 1.3 Hz, 1H), 5.79 (s, 1H), 6.30 (dd, *J* = 1.3, 1.3 Hz, 1H), 7.27–7.38 (m, 5H). ^13^C NMR (176 MHz, chloroform-*d*) δ 9.70, 28.87, 45.95, 70.56, 78.27, 123.10, 127.93 (2 × C), 128.18, 128.62 (2 × C), 139.55, 142.36, 197.64. EI-MS [M]^+^ = 216.0. Anal. Calcd for C_14_H_16_O_2_: C, 77.75; H, 7.46. Found: C, 77.44; H, 7.48.

*Trans-2,6-diisopropyl-3-methylidenetetrahydro-4H-pyran-4-one* (**13d**) (19.6 mg, 50%). Colorless oil. ^1^H NMR (700 MHz, chloroform-*d*) δ 0.87 (d, *J* = 6.7 Hz, 3H), 0.92 (d, *J* = 6.8 Hz, 3H), 0.99–1.04 (m, 6H), 1.75 (dhept, *J* = 13.5, 6.8 Hz, 1H), 1.91 (dhept, *J* = 9.6, 6.7 Hz, 1H), 2.35 (dd, *J* = 16.5, 10.6 Hz, 1H), 2.55 (dd, *J* = 16.5, 3.3 Hz, 1H), 3.68 (ddd, *J* = 10.6, 6.8, 3.3 Hz, 1H), 4.08 (ddd, *J* = 9.6, 1.3, 1.3 Hz, 1H), 5.12 (dd, *J* = 1.3, 1.3 Hz, 1H), 5.98 (dd, *J* = 1.3, 1.3 Hz, 1H). ^13^C NMR (176 MHz, chloroform-*d*) δ 18.28, 18.33, 18.56, 19.35, 29.34, 33.57, 44.03, 73.78, 83.03, 120.89, 144.18, 199.12. EI-MS [M]^+^ = 196.0. Anal. Calcd for C_12_H_20_O_2_: C, 73.43; H, 10.27. Found: C, 73.31; H, 10.24.

*Trans-2-butyl-6-isopropyl-3-methylidenetetrahydro-4H-pyran-4-one* (**13e**) (34.9 mg, 83%). Colorless oil. ^1^H NMR (700 MHz, chloroform-*d*) δ 0.89–0.93 (m, 6H), 0.99 (d, *J* = 6.7 Hz, 3H), 1.27–1.50 (m, 5H), 1.73 (hept, *J* = 6.7 Hz, 1H), 1.77–1.87 (m, 1H), 2.35 (dd, *J* = 16.8, 11.0 Hz, 1H), 2.53 (dd, *J* = 16.8, 3.0 Hz, 1H), 3.64 (ddd, *J* = 11.0, 6.7, 3.0 Hz, 1H), 4.60 (dddd, *J* = 10.0, 4.7, 1.5, 1.5 Hz, 1H), 4.98–5.39 (m, 1H), 6.02 (dd, *J* = 1.5, 1.5 Hz, 1H). ^13^C NMR (176 MHz, chloroform-*d*) δ 14.14, 18.28, 18.48, 22.48, 27.74, 33.24, 34.03, 43.66, 73.06, 76.34, 120.18, 145.35, 198.25. EI-MS [M]^+^ = 210.0. Anal. Calcd for C_13_H_22_O_2_: C, 74.24; H, 10.54. Found: C, 74.38; H, 10.50.

*Trans-6-isopropyl-3-methylidene-2-phenyltetrahydro-4H-pyran-4-one* (**13f**) (24.0 mg, 52%). Colorless oil. ^1^H NMR (700 MHz, chloroform-*d*) δ 0.84 (d, *J* = 6.8 Hz, 3H), 0.92 (d, *J* = 6.8 Hz, 3H), 1.67–1.79 (m, 1H), 2.43 (dd, *J* = 16.9, 10.9 Hz, 1H), 2.51 (dd, *J* = 16.9, 3.2 Hz, 1H), 3.42 (ddd, *J* = 10.9, 6.6, 3.2 Hz, 1H), 5.27 (dd, *J* = 1.3, 1.3 Hz, 1H), 5.81 (s, 1H), 6.31 (dd, *J* = 1.3, 1.3 Hz, 1H), 7.28–7.38 (m, 5H). ^13^C NMR (176 MHz, chloroform-*d*) δ 18.10, 18.25, 33.07, 43.75, 73.83, 78.36, 122.97, 128.06 (2 × C), 128.18, 128.58 (2 × C), 139.49, 142.24, 198.08. EI-MS [M]^+^ = 230.0. Anal. Calcd for C_15_H_18_O_2_: C, 78.23; H, 7.88. Found: C, 78.39; H, 7.87.

*Trans-2-isopropyl-3-methylidene-6-phenyltetrahydro-4H-pyran-4-one* (**13g**) (22.1 mg, 48%). Colorless oil. ^1^H NMR (700 MHz, chloroform-*d*) δ 0.94 (d, *J* = 6.6 Hz, 3H), 1.04 (d, *J* = 6.6 Hz, 3H), 1.99–2.10 (m, 1H), 2.76 (dd, *J* = 16.7, 9.8 Hz, 1H), 2.85 (dd, *J* = 16.7, 3.9 Hz, 1H), 4.22 (ddd, *J* = 9.0, 1.3, 1.3 Hz, 1H), 5.10 (dd, *J* = 9.8, 3.9 Hz, 1H), 5.20 (dd, *J* = 1.3, 1.3 Hz, 1H), 6.08 (dd, *J* = 1.3, 1.3 Hz, 1H), 7.28–7.34 (m, 1H), 7.35–7.41 (m, 4H). ^13^C NMR (176 MHz, chloroform-*d*) δ 17.97, 19.16, 29.70, 47.04, 70.92, 82.59, 121.33, 126.05 (2 × C), 128.03, 128.67 (2 × C), 140.65, 143.69, 197.78. EI-MS [M]^+^ = 230.0. Anal. Calcd for C_15_H_18_O_2_: C, 78.23; H, 7.88. Found: C, 78.21; H, 7.90.

*Trans-2-buthyl-3-methylidene-6-phenyltetrahydro-4H-pyran-4-one* (**13h**) (37.6 mg, 77%). Colorless oil. ^1^H NMR (700 MHz, chloroform-*d*) δ 0.91 (t, *J* = 7.2 Hz, 3H), 1.27–1.68 (m, 5H), 1.93 (dtd, *J* = 14.4, 9.5, 4.9 Hz, 1H), 2.76 (dd, *J* = 16.9, 10.0 Hz, 1H), 2.84 (dd, *J* = 16.9, 3.7 Hz, 1H), 4.66–4.72 (m, 1H), 5.10 (dd, *J* = 10.0, 3.7 Hz, 1H), 5.26 (dd, *J* = 1.3, 1.3 Hz, 1H), 6.11 (dd, *J* = 1.3, 1.3 Hz, 1H), 7.29–7.34 (m, 1H), 7.35–7.41 (m, 4H). ^13^C NMR (176 MHz, chloroform-*d*) δ 14.12, 22.52, 27.71, 34.04, 46.91, 70.46, 76.40, 120.66, 126.18 (2 × C), 128.14, 128.80 (2 × C), 140.79, 145.00, 197.23. EI-MS [M]^+^ = 244.0. Anal. Calcd for C_16_H_20_O_2_: C, 78.65; H, 8.25. Found: C, 78.33; H, 8.28.

*Trans-3-methylidene-2,6-diphenyltetrahydro-4H-pyran-4-one* (**13i**) (31.7 mg, 60%). Colorless oil. ^1^H NMR (700 MHz, chloroform-*d*) δ 2.85 (d, *J* = 7.0 Hz, 2H), 4.90 (dd, *J* = 7.0 Hz, 1H), 5.32 (s, 1H), 5.88 (s, 1H), 6.38 (s, 1H), 7.28–7.42 (m, 10H). ^13^C NMR (176 MHz, chloroform-*d*) δ 47.04, 71.29, 78.66, 123.59, 126.21 (2 × C), 128.06 (2 × C), 128.22, 128.48, 128.81 (2 × C), 128.84 (2 × C), 139.15, 140.53, 142.38, 197.05. EI-MS [M]^+^ = 264.0. Anal. Calcd for C_18_H_16_O_2_: C, 81.79; H, 6.10. Found: C, 81.92; H, 6.12.

*Trans-2-isopropyl-6-(4-methoxyphenyl)-3-methylidenetetrahydro-4H-pyran-4-one* (**13j**) (21.3 mg, 41%). Colorless oil. ^1^H NMR (700 MHz, chloroform-*d*) δ 0.94 (d, *J* = 6.6 Hz, 3H), 1.03 (d, *J* = 6.6 Hz, 3H), 2.04 (dhept, *J* = 8.9, 6.6 Hz, 1H), 2.77 (dd, *J* = 16.7, 9.1 Hz, 1H), 2.81 (dd, *J* = 16.7, 4.3 Hz, 1H), 3.81 (s, 3H), 4.18 (ddd, *J* = 8.9, 1.3, 1.3 Hz, 1H), 5.05 (dd, *J* = 9.1, 4.3 Hz, 1H), 5.18 (dd, *J* = 1.3, 1.3 Hz, 1H), 6.07 (d, *J* = 1.3, 1.3 Hz, 1H), 6.87–6.96 (m, 2H), 7.27–7.35 (m, 2H).^13^C NMR (176 MHz, chloroform-*d*) δ 18.06, 19.31, 29.91, 47.05, 55.46, 70.80, 82.46, 114.20 (2 × C), 121.34, 127.64 (2 × C), 132.82, 143.94, 159.55, 198.15. EI-MS [M]^+^ = 260.0. Anal. Calcd for C_16_H_20_O_3_: C, 73.82; H, 7.74. Found: C, 73.60; H, 7.75.

*Trans-2-butyl-6-(4-methoxyphenyl)-3-methylidenetetrahydro-4H-pyran-4-one* (**13k**) (45.0 mg, 82%). Colorless oil. ^1^H NMR (700 MHz, chloroform-*d*) δ 0.91 (t, *J* = 7.2 Hz, 3H), 1.29–1.51 (m, 4H), 1.59 (ddt, *J* = 14.1, 10.6, 5.3 Hz, 1H), 1.91 (dtd, *J* = 14.3, 9.6, 4.8 Hz, 1H), 2.77 (dd, *J* = 16.9, 9.3 Hz, 1H), 2.81 (dd, *J* = 16.9, 4.2 Hz, 1H), 3.81 (s, 3H), 4.64 (dddd, *J* = 9.4, 5.2, 1.5, 1.5 Hz, 1H), 5.05 (dd, *J* = 9.3, 4.2 Hz, 1H), 5.24 (dd, *J* = 1.3 Hz, 1.3 Hz, 1H), 6.10 (dd, *J* = 1.3 Hz, 1.3 Hz, 1H), 6.89–6.92 (m, 2H), 7.28–7.31 (m, 2H). ^13^C NMR (176 MHz, chloroform-*d*) δ 14.12, 22.52, 27.70, 34.05, 46.75, 55.43, 70.22, 76.13, 114.17 (2 × C), 120.53, 127.63 (2 × C), 132.79, 145.10, 159.52, 197.45. EI-MS [M]^+^ = 274.0. Anal. Calcd for C_17_H_22_O_3_: C, 74.42; H, 8.08. Found: C, 74.11; H, 8.11.

*Trans-6-(4-methoxyphenyl)-3-methylidene-2-phenyltetrahydro-4H-pyran-4-one* (**13l**) (29.4 mg, 50%). Colorless oil. ^1^H NMR (700 MHz, chloroform-*d*) δ 2.82 (dd, *J* = 16.9, 4.1 Hz, 1H), 2.86 (dd, *J* = 17.0, 9.7 Hz, 1H), 3.80 (s, 3H), 4.86 (dd, *J* = 9.7, 4.0 Hz, 1H), 5.29 (dd, *J* = 1.3, 1.3 Hz, 1H), 5.82 (dd, *J* = 1.3, 1.3 Hz, 1H), 6.36 (dd, *J* = 1.3, 1.3 Hz, 1H), 6.83–6.96 (m, 2H), 7.22–7.25 (m, 2H), 7.31–7.36 (m, 1H), 7.37–7.40 (m, 4H). ^13^C NMR (176 MHz, chloroform-*d*) δ 46.86, 55.46, 71.07, 78.48, 114.19 (2 × C), 123.51, 127.70 (2 × C), 128.06 (2 × C), 128.43, 128.82 (2 × C), 132.49, 139.30, 142.53, 159.59, 197.27. EI-MS [M]^+^ = 294.0. Anal. Calcd for C_19_H_18_O_3_: C, 77.53; H, 6.16. Found: C, 77.61; H, 6.15.

*Trans-6-ferrocenyl-2-isopropyl-3-methylidenetetrahydro-4H-pyran-4-one* (**13m**) (21.6 mg, 32%) Orange solid mp 136–138 °C. ^1^H NMR (700 MHz, chloroform-*d*) δ 0.92 (d, *J* = 6.7 Hz, 3H), 1.01 (d, *J* = 6.7 Hz, 3H), 1.99 (dp, *J* = 7.4, 6.6 Hz, 1H), 2.85 (dd, *J* = 16.9, 7.6 Hz, 1H), 2.95 (dd, *J* = 16.9, 4.8 Hz, 1H), 4.08–4.10 (m, 1H), 4.11 (dtd, *J* = 2.6, 1.3, 0.4 Hz, 1H), 4.15 (s, 5H), 4.16–4.18 (m, 1H), 4.21 (td, *J* = 2.5, 1.3 Hz, 1H), 4.28 (dt, *J* = 2.6, 1.3 Hz, 1H), 5.01 (dd, *J* = 7.6, 4.8 Hz, 1H), 5.17 (dd, *J* = 1.4, 1.4 Hz, 1H), 6.11 (dd, *J* = 1.4, 1.4 Hz, 1H). ^13^C NMR (176 MHz, chloroform-*d*) δ 17.41, 19.38, 30.33, 45.07, 66.65, 68.27, 68.40, 68.54, 68.87 (5 × C), 68.97, 80.51, 87.14, 121.17, 144.18, 198.26. EI-MS [M]^+^ = 338.0. Anal. Calcd for C_19_H_22_FeO_2_: C, 67.47; H, 6.56. Found: C, 67.66; H, 6.53.

*Trans-2-buthyl-6-ferrocenyl-3-methylidenetetrahydro-4H-pyran-4-one* (**13n**) (42.3 mg, 60%) Orange oil. ^1^H NMR (700 MHz, chloroform-*d*) δ 0.91 (t, *J* = 7.2 Hz, 3H), 1.19–1.51 (m, 4H), 1.60 (dddd, *J* = 14.0, 10.3, 5.5, 4.6 Hz, 1H), 1.79 (dtd, *J* = 14.1, 9.5, 4.6 Hz, 1H), 2.85 (dd, *J* = 17.0, 7.4 Hz, 1H), 2.94 (dd, *J* = 17.1, 4.9 Hz, 1H), 4.10 (dtd, *J* = 2.5, 1.3, 0.4 Hz, 1H), 4.15 (s, 5H), 4.16–4.18 (m, 1H), 4.21 (td, *J* = 2.5, 1.2 Hz, 1H), 4.27 (dt, *J* = 2.6, 1.3 Hz, 1H), 4.41–4.49 (m, 1H), 5.01 (dd, *J* = 7.4, 4.9 Hz, 1H), 5.21 (dd, *J* = 1.7, 1.1 Hz, 1H), 6.10 (dd, *J* = 1.8, 1.1 Hz, 1H). ^13^C NMR (176 MHz, chloroform-*d*) δ 14.17, 22.55, 27.52, 33.90, 44.96, 66.67, 68.26, 68.29, 68.40, 68.85 (5 × C), 68.94, 74.25, 87.11, 120.28, 145.40, 197.59. EI-MS [M]^+^ = 352.0. Anal. Calcd for C_20_H_24_FeO_2_: C, 68.19; H, 6.87. Found: C, 67.98; H, 6.90.

*Trans-6-ferrocenyl-3-methylidene-2-phenyltetrahydro-4H-pyran-4-one* (**13o**) (38.7 mg, 52%) Orange oil. ^1^H NMR (700 MHz, chloroform-*d*) δ 2.92 (dd, *J* = 17.1, 7.8 Hz, 1H), 2.97 (dd, *J* = 17.1, 4.8 Hz, 1H), 4.10 (s, 5H), 4.12–4.15 (m, 1H), 4.19 (td, *J* = 2.4, 1.3 Hz, 2H), 4.20–4.24 (m, 1H), 4.88 (dd, *J* = 7.7, 4.8 Hz, 1H), 5.14 (dd, *J* = 1.4, 1.4 Hz, 1H), 5.61 (dd, *J* = 1.6, 1.6 Hz, 1H), 6.32 (dd, *J* = 1.4, 1.4 Hz, 1H), 7.29–7.46 (m, 5H). ^13^C NMR (176 MHz, chloroform-*d*) δ 45.55, 66.51, 68.20, 68.35, 68.88 (6 × C), 69.14, 77.41, 87.00, 123.60, 128.14 (2 × C), 128.43, 128.74 (2 × C), 139.65, 143.27, 197.19. EI-MS [M]^+^ = 372.0. Anal. Calcd for C_22_H_20_FeO_2_: C, 70.99; H, 5.42. Found: C, 71.21; H, 5.41.

*2-(4-Methoxyphenyl)-5-methylidenetetrahydro-4H-pyran-4-one* (**15d**) (19.6 mg, 45%). Colorless oil. ^1^H NMR (700 MHz, chloroform-*d*) δ 2.71 (dd, *J* = 17.4, 11.1 Hz, 1H), 2.81 (dd, *J* = 17.4, 3.2 Hz, 1H), 3.81 (s, 3H), 4.57 (ddd, *J* = 14.5, 2.2, 2.2 Hz, 1H), 4.79 (ddd, *J* = 14.5, 1.2, 1.2 Hz, 1H), 4.79 (dd, *J* = 11.1, 3.2 Hz, 1H), 5.32 (ddd, *J* = 2.2, 1.2, 1.2 Hz, 1H), 6.15 (ddd, *J* = 2.2, 1.2, 1.2 Hz, 1H), 6.86–6.95 (m, 2H), 7.27–7.32 (m, 2H). ^13^C NMR (176 MHz, chloroform-*d*) δ 47.50, 55.46, 69.99, 77.67, 114.22 (2 × C), 120.55, 127.27 (2 × C), 132.70, 140.84, 159.63, 196.43. EI-MS [M]^+^ = 218.0. Anal. Calcd for C_13_H_14_O_3_: C, 71.54; H, 6.47. Found: C, 71.43; H, 6.49.

### 2.2. Biology


**Metabolic activity by MTT assay**


The MTT (3-(4,5-dimethyldiazol-2-yl)-2,5-diphenyl tetrazolium bromide) assay was performed to assess the cytotoxicity of the investigated compounds. Briefly, HL-60, MCF-7 and HUVEC cells were seeded in 24-well plates at density of 8 × 10^4^/mL. After 24 h incubation, cells were treated with various concentrations of the compounds and incubated for another 24 h or 48 h. Then, the cells were incubated with MTT solution (100 µL; 5 mg/mL in PBS) for 1.5 h and centrifuged. The formazan product was dissolved in DMSO, and the absorption was measured using FlexStation 3 Multi-Mode Microplate Reader (Molecular Devices, LLC, San Jose, CA, USA) at 560 nm. Assays were performed twice in triplicate for each analog.


**Cell morphology**


HL-60 cells were treated with compound **13d** or **13g** at IC_50_ and 2IC_50_ for 24 h. Then, the cells were examined for morphological changes using a light microscope at 40x magnification and photographed.


**Apoptosis assay**


Apoptotic cells were detected using FITC Annexin V Apoptosis Detection Kit I (BD Biosciences, San Jose, CA, USA). HL-60 cells (2.0 × 10^5^ cells/mL) seeded in 6-well plates and cultured overnight were treated with various concentrations of **13d** or **13g** for 24 h. Then, the cells were washed with PBS and resuspended in the binding buffer. After staining with FITC Annexin V and propidium iodide (15 min, room temperature) cells were analyzed by flow cytometry using CytoFlex and Kaluza Analysis Software v2 (Beckman Coulter Inc., Brea, CA, USA).


**Analysis of apoptosis, DNA damage and cell cycle**


Apoptosis, DNA damage, proliferation and cell cycle kinetics were investigated using the Apoptosis, DNA Damage and Cell Proliferation Kit (BD Biosciences, San Jose, CA, USA) according to the manufacturer’s protocol. Briefly, HL-60 cells (2.0 × 10^5^ cells/mL) were seeded in 6-well plates and left to grow for 24 h. Then, the cells were treated with **13d** or **13g** at IC_50_ and 2IC_50_ concentrations. After 24 h incubation, the cells were treated with BrdU solution for 8 h. Then, the cells were collected by centrifugation, fixed and permeabilized. To expose the incorporated BrdU, the cells were incubated with DNase (300 µg/mL in PBS) for 1 h at 37 °C. Afterwards, the cells were stained with fluorescent antibodies, including Anti-BrdU, Anti-Cleaved PARP (Asp214) and Anti-H2AX (pS139) for 20 min at room temperature. Total DNA for cell cycle analysis was stained using DAPI solution (1 µg/mL in staining buffer). Finally, the cells were analyzed by flow cytometry using CytoFLEX (Beckman Coulter, Inc., Brea, CA, USA) and the data analysis was performed with Kaluza Analysis Software v2 (Beckam Coulter, Inc., Brea, CA. USA).


**Topoisomerase IIα activity**


Topoisomerase IIα activity inhibition was assessed using the Human Topoisomerase II Decatenation Assay (Inspiralis, Norwich, UK). Briefly, kDNA was incubated with ATP, assay buffer (50 mM Tris-HCl pH 7.5, 125 mM NaCl, 10 mM MgCl_2_, 5 mM DTT, 100 µg/mL albumin), human topoisomerase II enzyme and various concentrations of the investigated compounds for 30 min at 37 °C. The reaction was stopped by addition of STEB (40% (*w/v*) sucrose, 100 mM Tris-HCl pH 8, 1 mM EDTA, 0.5 mg/mL bromophenol blue) followed by DNA extraction with chloroform/*iso*-amyl alcohol mixture (24:1). The samples were loaded onto a 1% agarose gel and run in a TBE buffer at 85 V for 65 min. Gels were stained with ethidium bromide (1 µg/mL) for 15 min, destained in water and visualized with a transilluminator (ChemiDoc MP Imaging System, Bio-Rad, Hercules, CA, USA).


**Statistical analysis**


All data are expressed as mean ±SEM. Statistical analyses were carried out using Prism 6.0 (GraphPad Software Inc., San Diego, CA, USA). Statistical significance was assessed using Student’s t-test (for comparison of two groups) or one-way ANOVA followed by a post-hoc multiple comparison Student–Newman–Keuls test (for comparison of three or more groups). **p* < 0.05, ***p* < 0.01 and ****p* < 0.001 were considered significant.

## 3. Results and Discussion

### 3.1. Chemistry

Addition of diethyl 2-oxopropylphosphonate **7** to aliphatic or aromatic aldehydes was performed using conditions described by Wada et al. [21]. Treatment of **7** [22] with 1.1 equivalents of sodium hydride at room temperature and next 1.1 equivalents of *n*-butyllithium solution (2,5 M in hexane) at the temperature range from –30 to –40 °C gave dianion **8**, which was reacted with 1.2 equivalents of aldehyde (Figure 1). After reaction, work-up and purification by column chromatography produced 4-hydroxy-2-oxoalkylphosphonates **9a-e** in good yields (Table 1). Conversion of **9a-e** into 3-diethoxyphosphoryldihydropyran-4-ones **11a-e** was realized performing the reaction with N,N-dimethylformamide dimethyl acetal (DMF-DMA, 3 equivalents) followed by treatment of the intermediate dimethylaminovinylphosphonates **10a-e** with 2.5 equivalents of BF_3_ · Et_2_O for 24 h, at room temperature. Surprisingly, cyclization of **9e** with ferrocenyl substituent, which was obtained by reaction of dianion **8** with ferrocenecarboxaldehyde, was inefficient in these conditions. Analysis of the ^31^P NMR spectra of the crude reaction mixture showed a number of signals which indicated degradation of the obtained compound. However, when the reaction was performed with three equivalents of DMF-DMA for 3 h at room temperature, without addition of BF_3_ · Et_2_O, the expected 3-diethoxyphosphoryl-6-ferrocenyldihydropyran-4-one **11e** was obtained. Attempts to purify **11a-e** by column chromatography failed due to their decomposition. However, inspection of ^1^H, ^31^P and ^13^C NMR spectra of these compounds showed that their purity can be estimated as 80–95%. Therefore, crude dihydropyran-4-ones **11a-e** were used in the next step.

With 3-diethoxyphosphoryldihydropyran-4-ones **11a-e** in hand, we decided to introduce various substituents next to the phosphoryl group by performing conjugate addition to these compounds (Figure 2). The reactions proceeded smoothly with Gilman reagents, Ph_2_CuLi and *n*-Bu_2_CuLi in the presence of TMSCl at −78 °C in THF, to give the expected 2,6-disubsttituted-3-diethoxyphosphoryltetrahydropyran-4-ones **12a–j** as mixtures of two diastereoisomers, *r-*2*-trans-*3-*trans*-6-**12a****–j** and r-2-*cis*-3-*trans*-6-**12a–j**, and enol forms, *trans*-enol-**12a–j**. Yields and isomer ratios are given in Table 2. It is worth noticing that in all cases, yields are significantly higher for the phenyl substituent than for the *n*-butyl one. Evidently, the phenyl Gilman reagent is a better nucleophile in these reactions than the *n*-butyl one.

Structures of the obtained 3-diethoxyphosphoryl-2,6-disubstitutedtetrahydropyran-4-ones **12a–j** were established using ^1^H, ^13^C and ^31^P NMR spectra, and mass spectrometry. Analysis of NMR spectra allowed us to assign the relative configurations and preferred conformations of the obtained isomers. Figure 2 shows characteristic coupling constants for *r*-2-*trans*-3-*trans*-6- and *r*-2-*cis*-3-*trans*-6-2-*n*-butyl-6-ethyl-3-diethoxyphosphoryltetrahydropyran-4-one **12a** taken from the NMR spectra of the mixture of both diastereoisomers. Large coupling constants *J*_H5–H6_ (10.9 Hz and 8.7 Hz) and small coupling constants *J*_H5′–H6_ (2.9 Hz and 4.0 Hz) indicate diaxial and axial-equatorial relationships of these protons, respectively, and prove the equatorial position of the ethyl group in both diastereoisomers. On the other hand, a large coupling constant *J*_C1′(n-Bu)-P_ = 13.6 Hz in *r*-2-*trans*-3-*trans*-6-**12a** proves the diaxial relation of *n*-butyl and phosphoryl groups, whereas *J*_C1′(n-Bu)-P_ ~ 0 Hz in r-2-*cis*-3-*trans*-6-**12a** indicates the axial-equatorial position of these groups. Signals of the enol form, *trans-*enol-**12a**, were also present, including a characteristic doublet of hydroxyl group with chemical shifts 11–12 ppm and coupling constant *J*_H–P_ ~1 Hz. This coupling constant shows the existence of resonance assisted hydrogen bond (RAHB) reported recently for the related organophosphorus compounds [23]. As similar sets of coupling constants were observed for 2,6-disubstituted 3-diethoxyphosphoryltetrahydropyran-4-ones **12b–j**, we believe that these assignments are valid for the whole series.

It is worth stressing that we did not observe the formation of the second pair of diastereoisomers, i.e., *r-*2*-trans-*3*-cis-6*- and *r-*2*-cis-*3*-cis-6*-**12a–j** or the corresponding *cis-*enol forms. This observation shows that the addition of Gilman reagents to 3-diethoxyphosphoryldihydropyran-4-ones **11a–e** is fully diastereoselective, and as expected, proceeds exclusively via axial attack of the Gilmann reagent on the dihydropyranone ring.

Unfortunately, the introduction of an isopropyl group in position 2 by addition of the appropriate Gilman reagent, *i*-Pr_2_CuLi, to 3-diethoxyphosphoryldihydropyran-4-ones **11a–e**, was inefficient. However, when we performed this addition using Grignard reagent, *i*-PrMgCl, the reaction proceeded smoothly and gave expected 6-substituted-3-diethoxyphosphoryl-2-isopropyltetrahydropyran-4-ones **12k–o** in good yields (Figure 3, Table 3). The only drawback was poor stereoselectivity of these reactions. In each case, a complex mixture of four diastereoisomers and two enol forms were formed. Although the axial attack was still favored, the substantial amount of *r*-2-*trans*-3-*cis*-6-**12k–o** along with their epimers, *r*-2-*cis*-3-*cis*-6-**12k–o,** and *cis*-enols-**12k–o**, were formed.

Figure 3 shows all six possible isomers of 3-diethoxyphosphoryl-2,6-diisopropylpyran-4-one **12l** and characteristic coupling constants which could be read from the NMR spectra. Despite many signals present in these spectra, we were able to assign signals and consequently relative configurations and conformations for five isomers in the mixture. The sixth isomer, *r*-2-*cis*-3-*trans*-6-**12l**, was present in too small an amount (1% according to ^31^P NMR spectrum, Table 3) to find its characteristic signals in ^1^H or ^13^C NMR spectra. Small values of coupling constants *J*_C1′-P_ for both *r*-2-*trans*-3-*cis*-6 and *r*-2-*cis*-3-*cis*-6-**12l** (3.5 Hz and 3.9 Hz, respectively) indicate a *cis* relationship between phosphoryl and isopropyl groups. In turn, small coupling constant J_H2-P_ in *r*-2-*trans*-3-*cis*-6-**12l** (11.8 Hz) and big coupling constant J_H2-P_ in *r*-2-*cis*-3-*cis*-6-**12l** (39.3 Hz) prove axial/equatorial and diaxial relationships between the H2 proton and phosphoryl group in these diastereoisomers, respectively. Similarities in the NMR spectra of **12l** and remaining 6-substituted 2-isopropyl-3-diethoxyphosphoryltetrahydropyran-4-ones **12k–o** allow us to propose the same stereochemical relationships in all pyranones **12k–o**.

Finally, the 2,6-disubstituted 3-diethoxyphosphoryltetrahydropyran-4-ones **12a–o** were converted to 3-methylidenetetrahydropyran-4-ones **13a–o** by performing Horner–Wadsworth–Emmons olefination. The reaction was carried out with 2 equivalents of K_2_CO_3_ and 10 equivalents of formaline at 0 °C in THF as a solvent (Figure 4). 2-*n*-Butyl- and 2-phenyl-3-methylidenetetrahydropyran-4-ones **13a–j** were formed as single *trans*-diastereoisomers, whereas 2-isopropyl-3-methylidenetetrahydropyran-4-ones **12k–o** were formed as mixtures of *trans*- and *cis*-diastereoisomers. However, diastereoisomers *cis*-**16k–o** were unstable and decomposed during column chromatography. Therefore, only *trans*-2,6-disubstituted 3-methylidenetetrahydropyran-4-ones **13a–o** were purified by column chromatography and obtained in moderate to very good yields (Table 4). Interestingly, all 2-*n*-butyl-3-methylidenetetrahydropyran-4-ones **13b,e,h,k,n** were obtained in significantly higher yields than 3-methylidenetetrahydropyran-4-ones with an isopropyl or phenyl group in position 2. It is very likely that greater steric hindrance of isopropyl or phenyl groups, in comparison to n-butyl, impedes HWE olefination of formaldehyde.

To broaden the substitution pattern of methylidenetetrahydropyran-4-ones, we subjected 3-diethoxyphosphoryldihydropyran-4-ones **11a–d** to a double bond reduction. A solution of lithium tri-sec-butylborohydride (L-Selectride^®^, 1M in THF) was used as a reducing agent, and the reaction was carried out in an inert atmosphere in the presence of 1.1 equivalents of reductant, with THF as a solvent at temperature range from −78 to 0 °C (Figure 5). After reaction work-up and purification by column chromatography, corresponding 2-substituted 5-diethoxyphosphoryltetrahydropyran-4-ones **14a–****d** were obtained with very good yields as mixtures of *trans*- and *cis*-diastereoisomers and enol form. (Table 5).

Relative configurations of the obtained compounds were assigned by the analysis of ^1^H and ^13^C NMR spectra, which showed that in all cases *trans* form dominated and the R^1^ group was in equatorial position. Characteristic coupling constants for **14c** are shown in Figure 4. Coupling constants of H-2 proton with vicinal protons H-3 and H-3′ (*J*_H2-H3_ = 10.8 or 11.6 Hz and *J*_H2-H3′_ = 2.9 or 3.4 Hz) indicate the equatorial positions of ethyl substituents in both diastereoisomers. On the other hand, the coupling constant of the H-5 proton with proton H-6 in *trans*-diastereoisomer (*J*_H5-H6_ = 10.8 Hz) indicate equatorial positions of the diethoxyphosphoryl group. In *cis*-diastereoisomer, the same coupling constant was significantly smaller (*J*_H5-H6_ = 4.2 Hz). Moreover, the coupling constant of H-6 proton with diethoxyphosphoryl group was very high (*J*_H6-P_ = 34.7 Hz), which indicates the axial position of diethoxyphosphoryl group. Similar sets of coupling constants were observed in other 2-substituted 5-diethoxyphosphoryltetrahydropyran-4-ones **14**.

Next, we made an attempt to transform the obtained 2-substituted-5-diethoxyphosphoryltetrahydropyran-4-ones **14a-d** into target 2-substituted-5-methylidenetetrahydropyran-4-ones **15a-d,** using Horner–Wadsworth–Emmons olefination. The reaction was carried out with 2 equivalents of K_2_CO_3_, 10 equivalents of formalin, THF as a solvent and room temperature for 3 h. Unfortunately, the obtained 3-methylidenetetrahydropyran-4-ones **15a-c,e**, turned out to be unstable at room temperature and decomposed during column chromatography. Only 2-(4-methoxyphenyl)-5-methylidenetetrahydropyran-4-one **15d** proved to be stable for a prolonged period of time (at least 7 days at room temperature) and after purification by column chromatography was obtained in 45% yield (Figure 6).

### 3.2. Biology

#### 3.2.1. In Vitro Cytotoxicity

The cytotoxic activity of the obtained methylidenetetrahydropyran-4-ones was investigated in two cancer cell lines, human promyelocytic leukemia (HL-60) and breast cancer adenocarcinoma (MCF-7), and in healthy umbilical vein endothelial cells (HUVEC), for comparison. An anticancer drug, carboplatin, was used as a reference compound. Metabolic activity of the cells was assessed after 48 h incubation with various concentrations of the analogs (Table 6). Purity of all final compounds was determined on the basis of their ^1^H NMR spectra [24] (signal-to-noise ratio and presence of unidentified signals) and was at least 98%, in most cases even over 99%.

Generally, the tested compounds were more cytotoxic for leukemic than for breast cancer cells. Analysis of the structure–activity relationship revealed that the cytotoxicity of methylidenetetrahydropyran-4-ones **13a–o** strongly depends on the nature of the substituent in position 2 (R^2^), and to a much lesser extent, on the structure of the substituent in position 6 (R^1^). This observation can be explained by the fact that R^2^ substituent is much closer to the methylidene group, which is believed to be the reaction site for various bionucleophiles, as mentioned in the Introduction. The best substituent in position 6 seems to be the isopropyl group; however, the differences in cytotoxic activity among other substituents in this position are not very significant. On the other hand, the presence and nature of the substituent next to the methylidene moiety is crucial for the activity. Compound **15d**, with no substituent in this position, is practically inactive. Evidently, the isopropyl group is the best substituent for position 2 in both tested cell lines. Almost all methylidenetetrahydropyran-4-ones **13** (with the exception of **13a**) bearing isopropyl in this position are highly cytotoxic, with IC_50_ values below 10 μM. The n-butyl or phenyl groups contribute much less to the cytotoxic activity of the analogs.

The most cytotoxic analogs were **13g**, **13j** and **13m,** with IC_50_ values below 4 µM against the HL-60 cell line. However, these compounds showed low cancer/healthy cell IC_50_ ratios, and therefore, poor selectivity for HL-60 cells compared to normal HUVEC cells. Analog **13d** showed high cytotoxicity in HL-60 cells (IC_50_ = 7.65 ± 0.05 µM) and was the most selective one, exhibiting over 3-fold higher inhibitory activity in HL-60 than in HUVEC cells.

Analogs **13d** and **13g** were selected for further assessment of their anticancer properties. The cytotoxicity of these two compounds in HL-60 cells was evaluated after 24 h incubation. The IC_50_ values were 15.65 ± 0.61 µM (Figure 5B) and 6.49 ± 0.31 µM (Figure 6B), respectively, and they were used in the subsequent assays.

#### 3.2.2. Antiproliferative Activity

To examine the influences of the investigated compounds on HL-60 cell proliferation, cells were exposed to a thymidine analog, bromodeoxyuridine (BrdU), that incorporates into newly synthesized DNA during proliferation. The results indicate that both analogs significantly inhibited cell proliferation. Treatment with **13d** at IC_50_ and 2IC_50_ concentrations decreased the BrdU-incorporating cell populations to 31.5 ± 1.65% and 26.4 ± 6.9%, respectively (Figure 5F), whereas **13g** at IC_50_ and 2IC_50_ concentrations decreased the populations to 35.3 ± 1.23% and 31.9 ± 6.02%, respectively (Figure 6F).

The cell cycle has a critical role in regulation of cell growth, proliferation and division of cells after DNA damage. Therefore, it often serves as a target in cancer therapy development [25]. The cell cycle distribution was investigated using DAPI DNA staining and flow cytometry analysis. **13d** at 2IC_50_ concentration significantly raised the number of cells in the G2/M phase (24.1 ± 1.40% of cells compared to control 14.9 ± 1.37%), whereas the population of cells in G0/G1 phase was diminished (Figure 5D,E). **13g** caused a slight increase in the G2/M cell population; however, the change was not statistically significant (Figure 6D,E).

#### 3.2.3. Cell Morphology Changes

HL-60 cells grow in suspension and usually have a roughly spherical shape. Treatment with **13d** or **13g** for 24 h induced several morphological changes in the cells. Some became elongated and asymmetrical; others exhibited characteristics typical for apoptotic cells, including shrinkage, membrane blebbing and formation of apoptotic bodies (Figure 5A and Figure 6A).

#### 3.2.4. Induction of Apoptosis

One of the typical features of apoptotic cell death is translocation of phosphatidylserine from the inside surface to the extracellular surface of the cell membrane [26], which can be detected with Annexin V [27]. Therefore, to examine the ability of the investigated compounds to induce apoptosis in HL-60 cells, we performed Annexin V and propidium iodide (PI) staining (Figure 5C and Figure 6C). In the control group, 90.3 ± 1.15% of HL-60 cells were healthy, 3.3 ± 0.36% were in early stage of apoptosis (Annexin V-positive and PI-negative) and 5.5 ± 1.02% were in late-stage apoptosis (Annaxin V-positive and PI-positive). **13d** did not induce a significant change in the number of apoptotic cells at IC_50_; however, at 2IC_50_, the analog caused increases in both early and late apoptotic cell numbers to 13.0 ± 2.64% and 24.6 ± 2.34%, respectively. Similarly, **13g** was not effective at IC_50_, but at 2IC_50_, early-stage apoptosis was detected in 8.7 ± 0.57% of cells, and 39.5 ± 1.16% of cells were in the late stage apoptosis.

Another hallmark of apoptosis is cleavage of poly(ADP-ribose) polymerase 1 (PARP1) into 24 kDa and 89 kDa fragments executed by caspases 3 and 7. PARP 1 fragmentation causes its inactivation, which in turn leads to inhibition of DNA repair and promotes caspase-mediated fragmentation of DNA during programmed cell death [28]. Therefore, detection of 89 kDa fragments of cleaved PARP with Anti-Cleaved PARP (Asp214) antibodies and flow cytometry was used to investigate apoptosis. Incubation of HL-60 cells with **13d** only at 2IC_50_ concentration led to significant increase in population of cells with cleaved PARP (49.0 ± 1.61%), in comparison to the control (0.7 ± 0.13%) (Figure 5G). Treatment with **13g** also caused a significant PARP cleavage in 13.5 ± 3.6% of the cell population only at 2IC_50_ concentration (Figure 6G).

#### 3.2.5. Induction of DNA Damage

DNA damage can be the result of either single- or double-strand breaks (DSBs). DSB are considered the most dangerous, as it is assumed that in metazoan, just one DSB has the potential to trigger cell death [29]. Cancer cells are often more susceptible to DSBs than healthy cells. Therefore, drugs able to induce DSB formation are widely used in anticancer therapies [30]. One of the earliest signals for DSB detection include phosphorylation of the histone H2AX on the Ser139, generating γH2AX [31]. In our study, induction of DSB was investigated by flow cytometry of HL-60 cells stained with anti-γH2AX fluorescent antibody. Treatment of HL-60 cells with **13d** at 2IC_50_ led to significant γH2AX generation in 38.4 ± 5.89% of cells in comparison to the control (1.1 ± 0.17%) [Figure 5H]. **13g** was markedly less genotoxic, causing at 2IC_50_, H2AX phosphorylation in 9.6 ± 2.94% of the cell population (Figure 6H).

#### 3.2.6. Topoisomerase IIα Activity

Some of the most effective DNA-damaging anticancer drugs include inhibitors of topoisomerases [32]. DNA topoisomerases are enzymes capable of changing DNA topology by generating transient breaks to mitigate strain. There are two types of topoisomerases, namely, type I enzymes cleaving one of the DNA strands and type II enzymes creating DSB [33]. Type II topoisomerases separate into subfamilies α and β. Topoisomerase IIβ is evenly distributed in all cells, both normal and cancerous, whereas topoisomerase IIα is overexpressed in rapidly proliferating cells, what makes it an optimal target for anticancer therapy [34]. Therefore, the inhibitory activity of the investigated analogs on topoisomerase IIα was also explored.

The ability of the analogs to inhibit topoisomerase II activity was investigated by measuring decatenation of kinetoplast DNA (kDNA) by human topoisomerase IIα. Topoisomerase II has the ability to decatenate interlinked dsDNA minicircles. When the enzyme is inactive, the kDNA molecules remain in the well after gel electrophoresis, whereas when topoisomerase II is active, the minicircles are released and migrate into the gel. Both compounds inhibited topoisomerase II activity at 0.5 mM and 1 mM concentrations. **13g** showed stronger inhibitory activity towards topoisomerase IIα than **13d** (Figure 7).

## 4. Conclusions

In the present investigation, we showed that Horner–Wadsworth–Emmons methodology is a very useful tool for the efficient synthesis of 2,6-disubstituted 3-methylidenetetrahydropyran-4-ones **13a–o** and 2-(4-methoxyphenyl)-5-methylidenetetrahydropyran-4-one **15d**. Addition of Gilman or Grignard reagents to intermediate diethoxyphosphoryltetrahydropyran-4-ones **13a-e** was fully or highly stereoselective, respectively, and finally gave access to diastereomerically pure *trans*-2,6-disubstituted 3-methylidenetetrahydropyran-4-ones **13a–o** with various substitution patterns. It is worth stressing that, in spite of very complex diastereoismeric mixtures of the obtained adducts, careful NMR analysis allowed us to establish their relative configurations and conformations. The in vitro assessment of the obtained analogs using the conventional MTT assay showed that in HL-60 and MCF-7 cells, analogs with isopropyl and phenyl substituents in position R2 were more cytotoxic than those with *n-*butyl substituents. Among the novel compounds, **13g** was the most cytotoxic and **13d** the most selective, as compared with normal HUVEC cells. Therefore, the anticancer potential of **13d** and **13g** in HL-60 cells was further investigated. Our findings showed that both analogs induced morphological changes characteristic of apoptosis in cancer cells, significantly inhibited proliferation and induced apoptotic cell death. Moreover, both compounds generated DNA damage, and analog **13d** arrested the cell cycle of HL-60 cells in the G2/M phase. It has been reported that topoisomerase II inhibitory activity is associated with generation of DNA damage and induction of cell cycle arrest at the G2/M phase [35]. Both **13d** and **13g** were able to inhibit the activity of topoisomerase IIα; however, **13g** was more potent. Based on these findings, the investigated analogs may be further optimized for the development of new and effective topoisomerase II inhibitors.

## Data Availability

Not applicable.

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
