# Peer review of "Stereoselective Synthesis and Anticancer Activity of 2,6-Disubstituted trans-3-Methylidenetetrahydropyran-4-ones"

_materials, 2022, doi:10.3390/ma15093030_

Round 1

Reviewer 1 Report

In this article by Bartosik et al. describes the stereoselective synthesis and anticancer activity of 2,6-disubstituted trans-3-methylidenetetrahydropyran-4-ones.

Please find my comments.

In scheme 1, di danion 8 has been generated by using two different bases. Can this reaction perform by using 2.5 equiv of any one base? All the chemistry involved here is well known in the literature.

Authors did a good job to establish the relative configurations and conformations by analyzing very complex diastereoismeric mixtures. It’s worth to mention that single crystal diffraction of one or two pure diastereoismer compounds could further confirm their configurations.

In vitro cytotoxicity reveals the poor selectivity of compounds for HL-60 cells versus normal HUVEC cells. Only compounds 13d showed some selectivity. So, practical usefulness is fewer in this series of compounds.

It will be very useful information if authors describe the mechanism of double-strand breaks (DSB) by using compounds 13g or 13d. Docking studies (intercalator or groove binder) might be helpful here.

Authors need to work more on this front to improve the biological applicability (efficient DNA cleaves or anticancer activity) of 13g or 13d. These two compounds are not sufficient for publication.

Reviewer 2 Report

This manuscript is devoted to a synthesis and cytotoxic activity of very interesting class of compounds and their derivatives. The authors successfully developed an effective and stereoselective method for the synthesis of 2,6-disubstituted-3-methylidenetetradydropyran-4-one with different substitution on the R1 and R2 positions. Two of their compounds 13g and 13d proved to be most cytotoxic and inhibited the activity of topoisomerase IIα.

The article is written at high level but I have a few comments and suggestions that will hopefully improve the manuscript if considered by the authors.

Authors characterized their compounds with NMR (1H and 13C) however some of the compounds which were mixtures (cis, trans, enol), proper assignment of signals was difficult to be assigned. Could 2D-NMR (NOESY/COSY) experiments be helpful to assign signals and differentiate compounds in a mixture?

How the purity of the compounds was determined? It may be good to provide %purity of final compounds used for cytotoxic activity?

In the R2 position of their pyran derivative, they introduced –isopropyl, n-butyl, and –phenyl substituents. In general discussion on the effect of these substituents on the synthesis and on the performance (cytotoxic activity) is missing in the present version of the manuscript. Perhaps a brief discussion about effect of these substituents on their cytotoxic activity may be helpful for future design of new drugs. For example, what are the general trends observed after substitution. Which substituent is best for anti-cancer activity and which position is favored most?

Furthermore, for the synthesis of 9a-e, phenyl substituent in the R1 position gave highest yield comparatively. Similarly, for the synthesis of 12a-j for introduction of R2 group, slightly higher yields were obtained with phenyl substituent in comparison to n-butyl ones. In contrast, for the synthesis of 13a-o, higher yields were obtained with n-butyl substituent in the R2 position in comparison to isopropyl and phenyl ones. Can authors explain briefly on these observations in the results and discussion section?

Finally, some typos observed;

On line 315, ‘N,N-dimethylamide dimethyl acetal’ (DMF-DMA) should be corrected to N,N-dimethylformamide dimethyl acetal.

In lines 482 and 494-503, Figures 2F, 3F, 2D-E, and 3D-E are mentioned while these Figures are missing. Perhaps its typo error.

In general, article is well written and can be published after minor revisions.

Reviewer 3 Report

The paper titled:

Stereoselective synthesis and anticancer activity of 2,6-disubsti-2 tuted trans-3-methylidenetetrahydropyran-4-ones.

Authors

Tomasz Bartosik , Joanna Drogosz-Stachowicz , Anna Janecka , Jacek Kędzia , Barbara Pacholczyk-Sienicka , Jacek Szymański , Katarzyna Gach-Janczak , Tomasz Janecki *

Manuscript ID

materials-1653665

Has been sent to Materials Journal (ISSN 1996-1944)

Section

Materials Chemistry

Special Issue

Research of Organic Molecules and Materials for Biological Application

The paper discusses the fowling

efficient and stereoselective synthesis of 2,6-disubstituted trans-3-methylidenetetrahydropyran-4-ones and 2-(4-methoxyphenyl)-5-methylidenetetrahydropyran-4-one that significantly broadens the spectrum of the available methylidenetetrahydropyran-4-ones with various substitution patterns. Target compounds were obtained using Horner-Wadsworth-Emmons methodology for the introduction of methylidene group onto the pyranone ring. 3-Diethoxyphosphoryltetrahydropyran-4-ones, which were key intermediates in this synthesis, were prepared by fully or highly stereoselective addition of Gilman or Grignard reagents to 3-diethoxyphosphoryldihydropyran-4-ones. Addition occurred preferentially by axial attack of the Michael donors on the dihydropyranone ring. Relative configurations and conformations of the obtained adducts were assigned using detailed analysis of the NMR spectra. The obtained methylidenepyran-4-ones were evaluated for the cytotoxic activity against two cancer cell lines (HL-60 and MCF-7). 2,6-Disubstituted 3-methylidenetetrahydropyran-4-ones with isopropyl and phenyl substituents in position 2 were more cytotoxic than analogs with n-butyl substituent. Two of the most cytotoxic analogs were then selected for further investigation on HL-60 cell line. Both analogs induced characteristic for apoptosis morphological changes in cancer cells, significantly inhibited proliferation and induced apoptotic cell death. Both compounds also generated DNA damage and one of the analogs arrested the cell cycle of HL-60 cells in the G2/M phase. In addition, both analogs were able to inhibit activity of topoisomerase IIα. Based on these findings, the investigated analogs may be further optimized for the development of new and effective topoisomerase II inhibitors.

Comments

1-         Some language corrections need

e.g., line 25 (detailed) should be (a detailed).

Line 26 (the cytotoxic) should be (cytotoxic) etc.

2-         Introduction

All the introductions spoke about biological activity there is nothing about synthesis.

3-Materials and Methods       

     (3.1. Chemistry) should be (2.1 chemistry).

About compounds from 7 to 12 there is nothing to indicate if these compounds are new or have been synthesized from previous or the authors synthesized them during this paper, no references nothing about them, but I found in the supporting information the method of synthesis and analysis of compounds 9a-e, 11a-e (general method only without analysis), 12a-j, 12k-o, 14a-d, I think if these compounds are new the methods should be in the paper, or if they are synthesized before the references should be in the paper, also the analysis should be completed for all new compounds.

4-         Compound 7, should write the company name that the authors purchased from as a reference.

5-         The nomenclature of compounds (13o) and (15d) should be revised with the nomenclature from the chem draw program.

6-         Lines 318,319,320 need more attention on how compound 11e was prepared, and what aldehyde was used.

7-         Is the toxicity of these compounds was measured and compared to their activity?

8-         References need to follow the journal instructions and need to revise and to but the publication site,

e.g. [CrossRef] [PubMed]

Reviewer 4 Report

Dear authors, 

see the attachment.

Round 2

Reviewer 1 Report

The revised manuscript failed to address my earlier comments about the smaller number of compounds (13g or 13d) showing little biological activities.

Authors should make this findings as a starting point to carry out further meaningful research. 

This manuscript is not ready for publication.
